# On the Variance of the Adaptive Learning Rate and Beyond

**Liyuan Liu** *
University of Illinois, Urbana-Champaign
`ll2@illinois`

**Haoming Jiang** †
Georgia Tech
`jianghm@gatech.edu`

**Pengcheng He, Weizhu Chen**
Microsoft Dynamics 365 AI
`{penhe,wzchen}@microsoft.com`

**Xiaodong Liu, Jianfeng Gao**
Microsoft Research
`{xiaodl,jfgao}@microsoft.com`

**Jiawei Han**
University of Illinois, Urbana-Champaign
`hanj@illinois`

## Abstract

The learning rate warmup heuristic achieves remarkable success in stabilizing training, accelerating convergence and improving generalization for adaptive stochastic optimization algorithms like RMSprop and Adam. Pursuing the theory behind warmup, we identify a problem of the adaptive learning rate – its variance is problematically large in the early stage, and presume warmup works as a variance reduction technique. We provide both empirical and theoretical evidence to verify our hypothesis. We further propose Rectified Adam (RAdam), a novel variant of Adam, by introducing a term to rectify the variance of the adaptive learning rate. Experimental results on image classification, language modeling, and neural machine translation verify our intuition and demonstrate the efficacy and robustness of RAdam.[1]

## 1 Introduction

Fast and stable optimization algorithms are what generations of researchers have been pursuing (Gauss, 1823; Cauchy, 1847). Remarkably, stochastic gradient-based optimization, such as stochastic gradient descent (SGD), has witnessed tremendous success in many fields of science and engineering despite its simplicity. Recently, many efforts have been made to accelerate optimization by applying *adaptive learning rate*. In particular, Adagrad (Duchi et al., 2010) and its variants, *e.g.*, RMSprop (Hinton et al., 2012), Adam (Kingma & Ba, 2014), Adadelta (Zeiler, 2012) and Nadam (Dozat, 2016), stand out due to their fast convergence, and have been considered as the optimizer of choice in many applications.

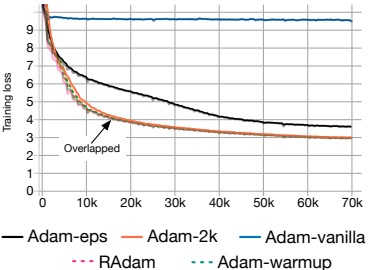

Figure 1: Training loss v.s. # of iterations of Transformers on the De-En IWSLT'14 dataset.

However, it has been observed that these optimization methods may converge to bad/suspicious local optima, and have to resort to a warmup heuristic – using a small learning rate in the first few epochs of training to mitigate such problem (Vaswani et al., 2017; Popel & Bojar, 2018). For example, when training typical Transformers based neural machine translation models on the De-En IWSLT'14 dataset, removing the warmup stage increases the training loss from 3 to around 10, as shown in Figure 1. Similar phenomena are observed in other scenarios like BERT (a bidirectional transformer language model) pre-training (Devlin et al., 2019).

Duo to the lack of the theoretical underpinnings, there is neither guarantee that warmup would bring consistent improvements for various machine learning settings nor guidance on how we should

---

*Work was done during an internship at Microsoft.

†Work was done during an internship at Microsoft.

[1]All implementations are available at: `https://github.com/LiyuanLucasLiu/RAdam`.

conduct warmup. Thus, researchers typically use different settings in different applications and have to take a trial-and-error approach, which can be tedious and time-consuming.

In this paper, we conduct both empirical and theoretical analysis of the convergence issue to identify its origin. We show that its root cause is: the adaptive learning rate has undesirably large variance in the early stage of model training, due to the limited amount of training samples being used. Thus, to reduce such variance, it is better to use smaller learning rates in the first few epochs of training, which justifies the warmup heuristic.

Inspired by our analysis results, we propose a new variant of Adam, called Rectified Adam (RAdam), which explicitly rectifies the variance of the adaptive learning rate based on derivations. We conduct extensive experiments on language modeling, image classification, and neural machine translation. RAdam brings consistent improvement over the vanilla Adam, which verifies the variance issue generally exists on various tasks across different network architectures.

In summary, our main contributions are two-fold:

- We identify the variance issue of the adaptive learning rate and present a theoretical justification for the warmup heuristic. We show that the convergence issue is due to the undesirably large variance of the adaptive learning rate in the early stage of model training.

- We propose a new variant of Adam (*i.e.*, RAdam), which not only explicitly rectifies the variance and is theoretically sound, but also compares favorably with the heuristic warmup.

## 2 PRELIMINARIES AND MOTIVATIONS

**Generic adaptive methods.** Algorithm 1 is a generic framework (all operations are element-wise). It describes various popular stochastic gradient descent algorithms (Reddi et al., 2018). Specifically, different optimization algorithms can be specified by different choices of $\phi(.)$ and $\psi(.)$, where $\phi(.)$ specifies how the momentum at time step $t$ is calculated, and $\psi(.)$ how the adaptive learning rate at $t$ is calculated. For example, in the Adam algorithm, we have:

$$\phi(g_1, \cdots, g_t) = \frac{(1 - \beta_1) \sum_{i=1}^{t} \beta_1^{t-i} g_t}{1 - \beta_1^t} \quad \text{and} \quad \psi(g_1, \cdots, g_t) = \sqrt{\frac{1 - \beta_2^t}{(1 - \beta_2) \sum_{i=1}^{t} \beta_2^{t-i} g_i^2}}. \quad (1)$$

For numerical stability, the function $\psi(.)$ in Equation 1 is usually calculated as $\widehat{\psi}(g_1, \cdots, g_t) = \frac{\sqrt{1 - \beta_2^t}}{\epsilon + \sqrt{(1 - \beta_2) \sum_{i=1}^{t} \beta_2^{t-i} g_i^2}}$, where $\epsilon$ is a relatively small / negligible value (*e.g.*, $1 \times 10^{-8}$).

---

**Algorithm 1:** Generic adaptive optimization method setup. All operations are element-wise.

**Input:** $\{\alpha_t\}_{t=1}^{T}$: step size, $\{\phi_t, \psi_t\}_{t=1}^{T}$: function to calculate momentum and adaptive rate,
$\quad\quad$ $\theta_0$: initial parameter, $f(\theta)$: stochastic objective function.
**Output:** $\theta_T$: resulting parameters
1 **while** $t = 1$ *to* $T$ **do**
2 $\quad$ $g_t \leftarrow \Delta_\theta f_t(\theta_{t-1})$ (Calculate gradients w.r.t. stochastic objective at timestep t)
3 $\quad$ $m_t \leftarrow \phi_t(g_1, \cdots, g_t)$ (Calculate momentum)
4 $\quad$ $l_t \leftarrow \psi_t(g_1, \cdots, g_t)$ (Calculate adaptive learning rate)
5 $\quad$ $\theta_t \leftarrow \theta_{t-1} - \alpha_t m_t l_t$ (Update parameters)
6 **return** $\theta_T$

---

**Learning rate warmup.** Instead of setting the learning rate $\alpha_t$ as a constant or in a decreasing order, a learning rate warmup strategy sets $\alpha_t$ as smaller values in the first few steps, thus not satisfying $\forall t \, \alpha_{t+1} \le \alpha_t$. For example, linear warmup sets $\alpha_t = t \, \alpha_0$ when $t < T_w$. Warmup has been demonstrated to be beneficial in many deep learning applications. For example, in the NMT experiments in Figure 1, the training loss convergences around 10 when warmup is not applied (Adam-vanilla), and it surprisingly decreases to below 3 after applying warmup (Adam-warmup).

To further analyze this phenomenon, we visualize the histogram of the absolute value of gradients on a log scale in Figure 2. We observe that, without applying warmup, the gradient distribution is distorted to have a mass center in relatively small values within 10 updates. Such gradient distortion means that the vanilla Adam is trapped in bad/suspicious local optima after the first few

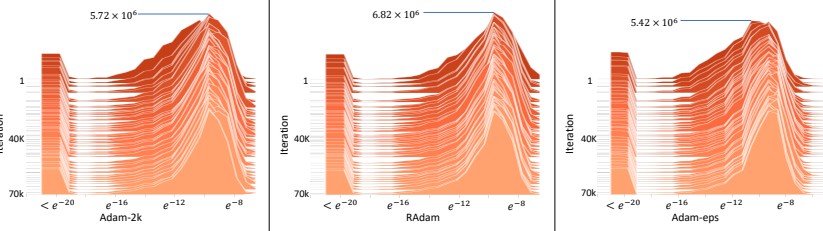

Figure 2: The absolute gradient histogram of the Transformers on the De-En IWSLT' 14 dataset during the training (stacked along the y-axis). X-axis is absolute value in the log scale and the height is the frequency. Without warmup, the gradient distribution is distorted in the first 10 steps.

Figure 3: The histogram of the absolute value of gradients (on a log scale) during the training of Transformers on the De-En IWSLT' 14 dataset. using Adam-2k, RAdam and Adam-eps.

updates. Warmup essentially reduces the impact of these problematic updates to avoid the convergence problem. In the following sections, we focus our analysis on learning rate warmup for the Adam algorithm, while it can be applied to other algorithms that use similar adaptive learning rate ($\psi(.)$) designs, *e.g.*, RMSprop (Hinton et al., 2012) and Nadam (Dozat, 2016).

## 3 VARIANCE OF THE ADAPTIVE LEARNING RATE

In this section, we first introduce empirical evidence, then analyze the variance of the adaptive learning rate to support our hypothesis – *Due to the lack of samples in the early stage, the adaptive learning rate has an undesirably large variance, which leads to suspicious/bad local optima*.

To convey our intuition, we begin with a special case. When $t = 1$, we have $\psi(g_1) = \sqrt{1/g_1^2}$. We view $\{g_1, \cdots, g_t\}$ as i.i.d. Gaussian random variables following $\mathcal{N}(0, \sigma^2)^2$. Therefore, $1/g_1^2$ is subject to the scaled inverse chi-squared distribution, Scale-inv-$\mathcal{X}^2(1, 1/\sigma^2)$, and $\mathrm{Var}[\sqrt{1/g_1^2}]$ is divergent. It means that the adaptive ratio can be undesirably large in the first stage of learning. Meanwhile, setting a small learning rate at the early stage can reduce the variance ($\mathrm{Var}[\alpha x] = \alpha^2 \mathrm{Var}[x]$), thus alleviating this problem. Therefore, we suggest it is the unbounded variance of the adaptive learning rate in the early stage that causes the problematic updates.

### 3.1 WARMUP AS VARIANCE REDUCTION

In this section, we design a set of controlled experiments to verify our hypothesis. Particularly, we design two variants of Adam that reducing the variance of the adaptive learning rate: *Adam-2k* and *Adam-eps*. We compare them to vanilla Adam with and without warmup on the IWSLT'14 German to English translation dataset (Cettolo et al., 2014).

In order to reduce the variance of the adaptive learning rate ($\psi(.)$), Adam-2k only updates $\psi(.)$ in the first two thousand iterations, while the momentum ($\phi(.)$) and parameters ($\theta$) are fixed[3]; other than this, it follows the original Adam algorithm. To make comparison with other methods, its iterations are indexed from -1999 instead of 1. In Figure 1, we observe that, after getting these additional two thousand samples for estimating the adaptive learning rate, Adam-2k avoids the convergence problem of the vanilla-Adam. Also, comparing Figure 2 and Figure 3, getting large enough samples prevents the gradient distribution from being distorted. These observations verify our hypothesis that the lack of sufficient data samples in the early stage is the root cause of the convergence issue.

---

[2]The mean zero normal assumption is valid at the beginning of the training, since weights are sampled from normal distributions with mean zero (Balduzzi et al., 2017), further analysis is conducted in Section 5.3.

[3]Different from Gotmare et al. (2019), all parameters and first moments are frozen in the first 2000 iterations.

Another straightforward way to reduce the variance is to increase the value of $\epsilon$ in $\widehat{\psi}(g_1, \cdots, g_t) = \frac{\sqrt{1-\beta_2^t}}{\epsilon + \sqrt{(1-\beta_2)\sum_{i=1}^t \beta_2^{t-i} g_i^2}}$. Actually, if we assume $\widehat{\psi}(.)$ is subject to the uniform distribution, its variance equals to $\frac{1}{12\epsilon^2}$. Therefore, we design Adam-eps, which uses a non-negligibly large $\epsilon = 10^{-4}$, while $\epsilon = 10^{-8}$ for vanilla Adam. Its performance is summarized in Figure 1. We observe that it does not suffer from the serious convergence problem of vanilla-Adam. This further demonstrates that the convergence problem can be alleviated by reducing the variance of the adaptive learning rate, and also explains why tuning $\epsilon$ is important in practice (Liu et al., 2019). Besides, similar to Adam-2k, it prevents the gradient distribution from being distorted (as shown in Figure 3). However, as in Figure 1, it produces a much worse performance comparing to Adam-2k and Adam-warmup. We conjecture that this is because large $\epsilon$ induces a large bias into the adaptive learning rate and slows down the optimization process. Thus, we need a more principled and rigorous way to control the variance of the adaptive learning rate. In the next subsection, we will present a theoretical analysis of the variance of the adaptive learning rate.

## 3.2 ANALYSIS OF ADAPTIVE LEARNING RATE VARIANCE

As mentioned before, Adam uses the exponential moving average to calculate the adaptive learning rate. For gradients $\{g_1, \cdots, g_t\}$, their exponential moving average has a larger variance than their simple average. Also, in the early stage ($t$ is small), the difference of the exponential weights of $\{g_1, \cdots, g_t\}$ is relatively small (up to $1 - \beta_2^{t-1}$). Therefore, for ease of analysis, we approximate the distribution of the exponential moving average as the distribution of the simple average (Nau, 2014), i.e., $p(\psi(.)) = p(\sqrt{\frac{1-\beta_2^t}{(1-\beta_2)\sum_{i=1}^t \beta_2^{t-i} g_i^2}}) \approx p(\sqrt{\frac{t}{\sum_{i=1}^t g_i^2}})$. Since $g_i \sim \mathcal{N}(0, \sigma^2)$, we have $\frac{t}{\sum_{i=1}^t g_i^2} \sim$ Scale-inv-$\mathcal{X}^2(t, \frac{1}{\sigma^2})$. Therefore, we assume $\frac{1-\beta_2^t}{(1-\beta_2)\sum_{i=1}^t \beta_2^{t-i} g_i^2}$ also subjects to a scaled inverse chi-square distribution with $\rho$ degrees of freedom (further analysis on this approximation is conducted in Section 5.3). Based on this assumption, we can calculate $\text{Var}[\psi^2(.)]$ and the PDF of $\psi^2(.)$. Now, we proceed to the analysis of its square root variance, i.e., $\text{Var}[\psi(.)]$, and show how the variance changes with $\rho$ (which corresponds to number of used training samples).

**Theorem 1.** If $\psi^2(.) \sim$ Scale-inv-$\mathcal{X}^2(\rho, \frac{1}{\sigma^2})$, $\text{Var}[\psi(.)]$ monotonically decreases as $\rho$ increases.

*Proof.* For $\forall \rho > 4$, we have:

$$\text{Var}[\psi(.)] = \mathbb{E}[\psi^2(.)] - \mathbb{E}[\psi(.)]^2 = \tau^2 \left( \frac{\rho}{\rho-2} - \frac{\rho \, 2^{2\rho-5}}{\pi} \mathcal{B}(\frac{\rho-1}{2}, \frac{\rho-1}{2})^2 \right), \qquad (2)$$

where $\mathcal{B}(.)$ is the beta function. By analyzing the derivative of $\text{Var}[\psi(.)]$, we know it monotonically decreases as $\rho$ increases. The detailed derivation is elaborated in the Appendix A. $\qquad\square$

Theorem 1 gives a qualitative analysis of the variance of the adaptive learning rate. It shows that, due to the lack of used training samples in the early stage, $\text{Var}[\psi(.)]$ is larger than the late stage (Figure 8). To rigorously constraint the variance, we perform a quantified analysis on $\text{Var}[\psi(.)]$ by estimating the degree of freedoms $\rho$.

## 4 RECTIFIED ADAPTIVE LEARNING RATE

In the previous section, Equation 2 gives the analytic form of $\text{Var}[\psi(.)]$, where $\rho$ is the degree of freedoms. Here, we first give an estimation of $\rho$ based on $t$ to conduct a quantified analysis for $\text{Var}[\psi(g_1, \cdots, g_t)]$, then we describe the design of the learning rate rectification, and compare it to the heuristic warmup strategies.

## 4.1 ESTIMATION OF $\rho$

The exponential moving average (EMA) can be interpreted as an approximation to the simple moving average (SMA) in real application (Nau, 2014), i.e.,

$$p\left( \frac{(1-\beta_2) \sum_{i=1}^t \beta_2^{t-i} g_i^2}{1 - \beta_2^t} \right) \approx p\left( \frac{\sum_{i=1}^{f(t,\beta_2)} g_{t+1-i}^2}{f(t, \beta_2)} \right). \qquad (3)$$

---

**Algorithm 2:** Rectified Adam. All operations are element-wise.

---

**Input:** $\{\alpha_t\}_{t=1}^T$: step size, $\{\beta_1, \beta_2\}$: decay rate to calculate moving average and moving 2nd
moment, $\theta_0$: initial parameter, $f_t(\theta)$: stochastic objective function.

**Output:** $\theta_t$: resulting parameters

1   $m_0, v_0 \leftarrow 0, 0$ (Initialize moving 1st and 2nd moment)

2   $\rho_\infty \leftarrow 2/(1 - \beta_2) - 1$ (Compute the maximum length of the approximated SMA)

3   **while** $t = \{1, \cdots, T\}$ **do**

4      $g_t \leftarrow \Delta_\theta f_t(\theta_{t-1})$ (Calculate gradients w.r.t. stochastic objective at timestep t)

5      $v_t \leftarrow 1/\beta_2 v_{t-1} + (1 - \beta_2)g_t^2$ (Update exponential moving 2nd moment)

6      $m_t \leftarrow \beta_1 m_{t-1} + (1 - \beta_1)g_t$ (Update exponential moving 1st moment)

7      $\widehat{m_t} \leftarrow m_t/(1 - \beta_1^t)$ (Compute bias-corrected moving average)

8      $\rho_t \leftarrow \rho_\infty - 2t\beta_2^t/(1 - \beta_2^t)$(Compute the length of the approximated SMA)

9      **if** *the variance is tractable, i.e., $\rho_t > 4$* **then**

10        $l_t \leftarrow \sqrt{(1 - \beta_2^t)/v_t}$ (Compute adaptive learning rate)

11        $r_t \leftarrow \sqrt{\frac{(\rho_t-4)(\rho_t-2)\rho_\infty}{(\rho_\infty-4)(\rho_\infty-2)\rho_t}}$ (Compute the variance rectification term)

12        $\theta_t \leftarrow \theta_{t-1} - \alpha_t r_t \widehat{m_t} l_t$ (Update parameters with adaptive momentum)

13      **else**

14        $\theta_t \leftarrow \theta_{t-1} - \alpha_t \widehat{m_t}$ (Update parameters with un-adapted momentum)

15   **return** $\theta_T$

---

where $f(t, \beta_2)$ is the length of the SMA which allows the SMA to have the same "center of mass" with the EMA. In other words, $f(t, \beta_2)$ satisfies:

$$\frac{(1 - \beta_2)\sum_{i=1}^t \beta_2^{t-i} \cdot i}{1 - \beta_2^t} = \frac{\sum_{i=1}^{f(t,\beta_2)}(t + 1 - i)}{f(t, \beta_2)}. \tag{4}$$

By solving Equation 4, we have: $f(t, \beta_2) = \frac{2}{1-\beta_2} - 1 - \frac{2t\beta_2^t}{1-\beta_2^t}$. In the previous section, we assume: $\frac{1-\beta_2^t}{(1-\beta_2)\sum_{i=1}^t \beta_2^{t-i}g_i^2} \sim$ Scale-inv-$\mathcal{X}^2(\rho, \frac{1}{\sigma^2})$. Here, since $g_i \sim \mathcal{N}(0, \sigma^2)$, we have $\frac{\sum_{i=1}^{f(t,\beta_2)} g_{t+1-i}^2}{f(t,\beta_2)} \sim$ Scale-inv-$\mathcal{X}^2(f(t, \beta_2), \frac{1}{\sigma^2})$. Thus, Equation 3 views Scale-inv-$\mathcal{X}^2(f(t, \beta_2), \frac{1}{\sigma^2})$ as an approximation to Scale-inv-$\mathcal{X}^2(\rho, \frac{1}{\sigma^2})$. Therefore, we treat $f(t, \beta_2)$ as an estimation of $\rho$. For ease of notation, we mark $f(t, \beta_2)$ as $\rho_t$. Also, we refer $\frac{2}{1-\beta_2} - 1$ as $\rho_\infty$ (maximum length of the approximated SMA), due to the inequality $f(t, \beta_2) \leq \lim_{t\to\infty} f(t, \beta_2) = \frac{2}{1-\beta_2} - 1$.

## 4.2   VARIANCE ESTIMATION AND RECTIFICATION

Based on previous estimations, we have $\text{Var}[\psi(.)] = \tau^2(\frac{\rho_t}{\rho_t-2} - \frac{\rho_t 2^{2\rho_t-5}}{\pi}\mathcal{B}(\frac{\rho_t-1}{2}, \frac{\rho_t-1}{2})^2)$. The value of this function in the early stage is significantly larger than the late stage (as analyzed later, it decays roughly at the speed of $O(\frac{1}{\rho_t})$). For example, the variance at $\rho_t = 5$ is over 100 times larger than the variance at $\rho_t = 500$. Additionally, based on Theorem 1, we know $\min_{\rho_t} \text{Var}[\psi(.)] = \text{Var}[\psi(.)]|_{\rho_t=\rho_\infty}$ and mark this minimal value as $C_{\text{var}}$. In order to ensure that the adaptive learning rate ($\psi(.)$) has consistent variance, we rectify the variance at the $t$-th timestamp as below,

$$\text{Var}[r_t \, \psi(g_1, \cdots, g_t)] = C_{\text{var}} \quad \text{where} \quad r_t = \sqrt{C_{\text{var}}/\text{Var}[\psi(g_1, \cdots, g_t)]}.$$

Although we have the analytic form of $\text{Var}[\psi(.)]$ (*i.e.*, Equation 2), it is not numerically stable. Therefore, we use the first-order approximation to calculate the rectification term. Specifically, by approximating $\sqrt{\psi^2(.)}$ to the first order (Wolter, 2007),

$$\sqrt{\psi^2(.)} \approx \sqrt{\mathbb{E}[\psi^2(.)]} + \frac{1}{2\sqrt{\mathbb{E}[\psi^2(.)]}}(\psi^2(.) - \mathbb{E}[\psi^2(.)]) \quad \text{and} \quad \text{Var}[\psi(.)] \approx \frac{\text{Var}[\psi^2(.)]}{4\,\mathbb{E}[\psi^2(.)]}.$$

Since $\psi^2(.) \sim$ Scale-inv-$\mathcal{X}^2(\rho_t, \frac{1}{\sigma^2})$, we have:

$$\text{Var}[\psi(.)] \approx \rho_t/[2(\rho_t - 2)(\rho_t - 4)\sigma^2]. \tag{5}$$

In Section 5.3, we conduct simulation experiments to examine Equation 5 and find that it is a reliable approximation. Based on Equation 5, we know that $\text{Var}[\sqrt{\psi(.)}]$ decreases approximately at the

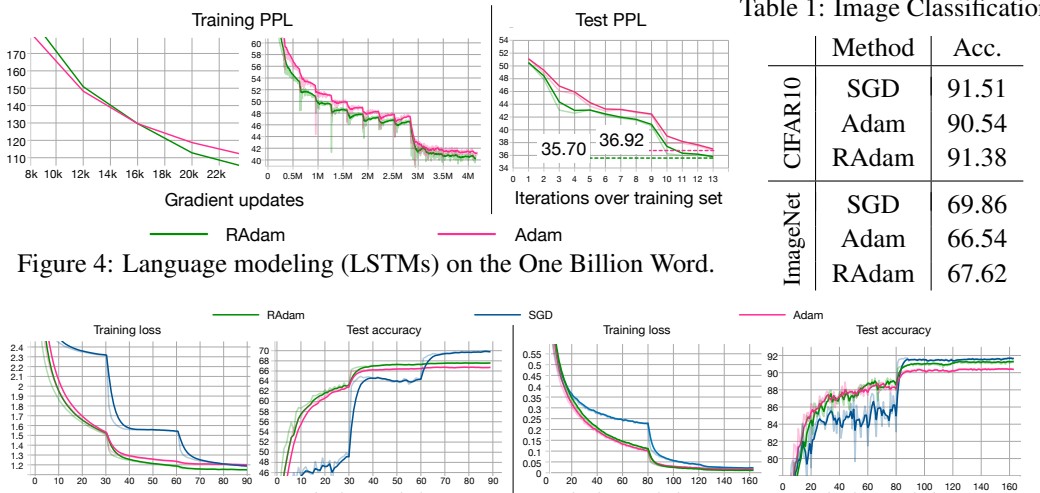

Figure 4: Language modeling (LSTMs) on the One Billion Word.

Table 1: Image Classification

| | Method | Acc. |
|---|---|---|
| CIFAR10 | SGD | 91.51 |
| | Adam | 90.54 |
| | RAdam | 91.38 |
| ImageNet | SGD | 69.86 |
| | Adam | 66.54 |
| | RAdam | 67.62 |

Figure 5: Training of ResNet-18 on the ImageNet and ResNet-20 on the CIFAR10 dataset.

speed of $O(\frac{1}{\rho_t})$. With this approximation, we can calculate the rectification term as:

$$r_t = \sqrt{\frac{(\rho_t - 4)(\rho_t - 2)\rho_\infty}{(\rho_\infty - 4)(\rho_\infty - 2)\rho_t}}.$$

Applying our rectification term to Adam, we come up with a new variant of Adam, Rectified Adam (RAdam), as summarized in Algorithm 2. Specifically, when the length of the approximated SMA is less or equal than 4, the variance of the adaptive learning rate is intractable and the adaptive learning rate is inactivated. Otherwise, we calculate the variance rectification term and update parameters with the adaptive learning rate. It is worth mentioning that, if $\beta_2 \leq 0.6$, we have $\rho_\infty \leq 4$ and RAdam is degenerated to SGD with momentum.

### 4.3 IN COMPARISON WITH WARMUP AND OTHER STABILIZATION TECHNIQUES

Different from the analysis in this paper, warmup is originally proposed to handle training with very large batches for SGD (Goyal et al., 2017; Gotmare et al., 2019; Bernstein et al., 2018; Xiao et al., 2017). We notice that $r_t$ has a similar form to the heuristic linear warmup, which can be viewed as setting the rectification term as $\frac{min(t, T_w)}{T_w}$. It verifies our intuition that warmup works as a variance reduction technique. RAdam deactivates the adaptive learning rate when its variance is divergent, thus avoiding undesired instability in the first few updates. Besides, our method does not require an additional hyperparameter (*i.e.*, $T_w$) and can automatically adapt to different moving average rules.

Here, we identify and address an underlying issue of adaptive optimization methods independent of (neural) model architectures. Thus, the proposed rectification term is orthogonal to other training stabilization techniques such as gradient clipping (Bengio et al., 2013), smoothing the adaptive learning rate (*i.e.*, increasing $\epsilon$, applying geometric mean filter (Chen & Gu, 2018), or adding range constraints (Luo et al., 2019)), initialization (Balduzzi et al., 2017; Zhang et al., 2019) and normalization (Ba et al., 2016; Ioffe & Szegedy, 2015). Indeed, these techniques can be combined with the proposed variance rectification method.

## 5 EXPERIMENTS

We evaluate RAdam on several benchmarks: One Billion Word for language modeling; Cifar10 and ImageNet for image classification; IWSLT'14 De-En/EN-DE and WMT'16 EN-De for neural machine translation. Following Loshchilov & Hutter (2018), we decouple weight decays in the vanilla Adam, Adam with warmup and RAdam in our experiments. Details are in Appendix B.

### 5.1 COMPARING TO VANILLA ADAM

As analyzed before, the adaptive learning rate has undesirably large variance in the early stage of training and leads to suspicious/bad local optima on NMT. One question we are interested in

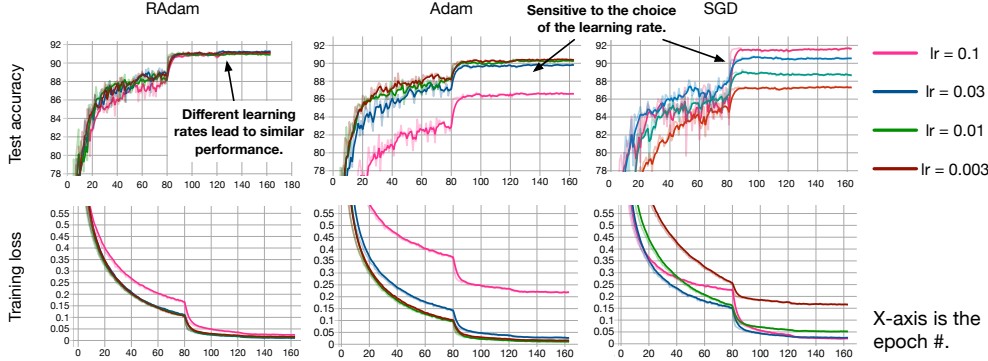

Figure 6: Performance of RAdam, Adam and SGD with different learning rates on CIFAR10.

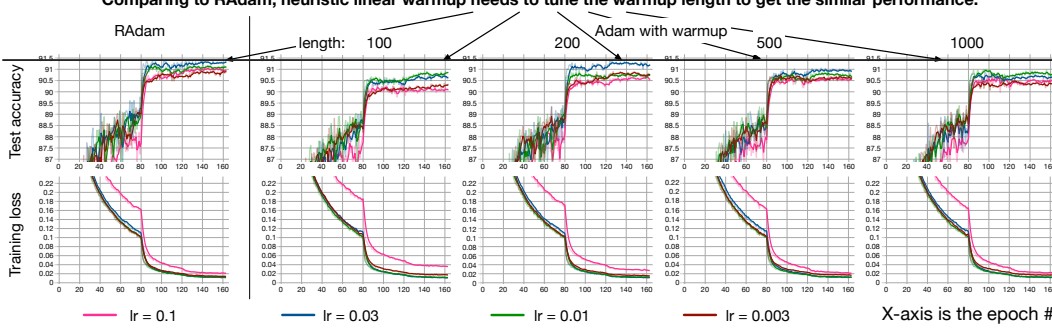

Figure 7: Performance of RAdam, Adam with warmup on CIFAR10 with different learning rates.

is: whether such an issue widely exits in other similar tasks and applications. Thus, we conduct a set of experiments with two classical tasks of NLP and CV, *i.e.*, language modeling and image classification. RAdam not only results in consistent improvements over the vanilla Adam, but also demonstrates its robustness to the change of learning rates. It verifies that the variance issue exists in various machine learning applications, and has a big impact on the model behavior.

**Performance Comparison.** The performances on language modeling (*i.e.*, One Billion Word (Chelba et al., 2013)) and image classification (*i.e.*, CIFAR10 (Krizhevsky et al., 2009) and ImageNet (Deng et al., 2009)) are presented in Figure 4, 5. The results show that RAdam outperforms Adam in all three datasets. As shown in Figure 4, although the rectification term makes RAdam slower than the vanilla Adam in the first few epochs, it allows RAdam to converge faster after that. In other words, by reducing the variance of the adaptive learning rate in the early stage, it gets both faster convergence and better performance, which verifies the impact of the variance issue. We also observe that RAdam obtains consistent improvements over Adam on image classification. It is worth noting that, on both ImageNet and CIFAR10, although RAdam fails to outperform SGD in terms of test accuracy, it results in a better training performance (*e.g.*, the training accuracy of SGD, Adam, and RAdam on ImageNet are 69.57, 69.12 and 70.30 respectively).

**Robustness to Learning Rate Change.** Besides performance improvements, RAdam also improves the robustness of model training. We use different initial learning rates, conduct experiments with ResNet-20 on the CIFAR10 datasets, and summarize their performance in Figure 6. For learning rates within a broad range (*i.e.*, $\{0.1, 0.03, 0.01, 0.003\}$), RAdam achieves consistent model performances (their test accuracy curves highly overlap with each other), while Adam and SGD are shown to be more sensitive to the learning rate. The observation can be interpreted that by rectifying the variance of the adaptive learning rate, RAdam improves the robustness of model training and can adapt to different learning rates of a broader range.

## 5.2 COMPARING TO HEURISTIC WARMUP

To examine the effectiveness of RAdam, we first conduct comparisons on neural machine translation, on which the state-of-the-art employs Adam with the linear warmup. Specifically, we conduct experiments on three datasets, i.e., IWSLT'14 De-En, IWSLT'14 En-De, and WMT'16 En-De. Due

Table 2: BLEU score on Neural Machine Translation.

| Method | IWSLT'14 DE-EN | IWSLT'14 EN-DE | WMT'16 EN-DE |
|---|---|---|---|
| Adam with warmup | $34.66 \pm 0.014$ | $28.56 \pm 0.067$ | 27.03 |
| RAdam | $34.76 \pm 0.003$ | $28.48 \pm 0.054$ | 27.27 |

to the limited size of the IWSLT'14 dataset, we conduct experiments using 5 different random seeds and report their mean and standard derivation. As discussed before, the vanilla Adam algorithm leads to suspicious/bad local optima (i.e., converges to a training perplexity around 500), and needs a learning rate warmup stage to stabilize the training.

We summarize the performance obtained with the heuristic warmup and our proposed rectification term in Table 2 and visualize the training curve of IWSLT De-En in Figure 1. With a consistent adaptive learning rate variance, our proposed method achieves similar performance to that of previous state-of-the-art warmup heuristics. It verifies our intuition that the problematic updates of Adam are indeed caused by the undesirably large variance in the early stage.

Moreover, we applied Adam with warmup on the CIFAR10 dataset. Its best accuracy on the test set is 91.29, which is similar to RAdam (91.38). However, we found that RAdam requires less hyperparameter tuning. Specifically, we visualize their learning curves in Figure 7. For some warmup steps, Adam with warmup is relatively more sensitive to the choice of the learning rate. RAdam, at the same time, is not only more robust, but also can automatically control the warmup behavior (*i.e.*, without requiring the length of warmup). For example, when setting the learning rate as 0.1, Adam with 100 steps of warmup fails to get satisfying performance and only results in an accuracy of 90.13; RAdam successfully gets an accuracy of 91.06, with the original setting of the moving average calculation (*i.e.*, $\beta_1 = 0.9, \beta_2 = 0.999$). We conjecture the reason is due to the fact that RAdam, which is based on a rigorous variance analysis, explicitly avoids the extreme situation where the variance is divergent, and rectifies the variance to be consistent in other situations.

### 5.3 SIMULATED VERIFICATION

In Sections 3 and 4, we approximate $\mathrm{Var}[\sqrt{t/\sum_{i=1}^t g_i^2}]$ to the first order, and assume $\psi^2(.) = \frac{1-\beta_2^t}{(1-\beta_2)\sum_{i=1}^t \beta_2^{t-i} g_i^2}$ subjects to a scaled inverse chi-square distribution (this assumption covers the approximation from EMA to SMA). Here, we examine these two approximations using simulations.

**First Order Approximation of** $\mathrm{Var}[\sqrt{t/\sum_{i=1}^t g_i^2}]$. To compare Equations 5 and 2, we assume $\tau = 1$ and plot their values and difference for $\nu = \{5, \cdots, 500\}$ in Figure 8. The curve of the analytic form and the first-order approximation highly overlap, and their difference is much smaller than their value. This result verifies that our first-order approximation is very accurate.

**Scaled Inverse Chi-Square Distribution Assumption.** In this paper, we assume $g_i$ accords to a Normal distribution with a zero mean. We also assume $\psi^2(.)$ accords to the scaled inverse chi-square distribution to derive the variance of $\mathrm{Var}[\psi(.)]$, based on the similarity between the exponential moving average and simple moving average. Here, we empirically verify this assumption.

Specifically, since $g_i$ in the optimization problem may not be zero-mean, we assume its expectation is $\mu$ and sample $g_i$ from $\mathcal{N}(\mu, 1)$. Then, based on these samples, we calculate the variance of the original adaptive learning rate and the proposed rectified adaptive learning rate, *i.e.*, $\mathrm{Var}[\frac{1}{\hat{v}_t}]$ and $\mathrm{Var}[\frac{r_t}{\hat{v}_t}]$ respectively. We set $\beta_2$ to 0.999, the number of sampled trajectories to 5000, the number of iterations to 6000, and summarize the simulation results in Figure 9. Across all six settings with different $\mu$, the adaptive learning rate has a larger variance in the first stage and the rectified adaptive learning rate has relative consistent variance. This verifies the reliability of our assumption.

## 6 CONCLUSION

In this paper, we explore the underlying principle of the effectiveness of the warmup heuristic used for adaptive optimization algorithms. Specifically, we identify that, due to the limited amount of samples in the early stage of model training, the adaptive learning rate has an undesirably large variance and can cause the model to converge to suspicious/bad local optima. We provide both empirical and theoretical evidence to support our hypothesis, and further propose a new variant

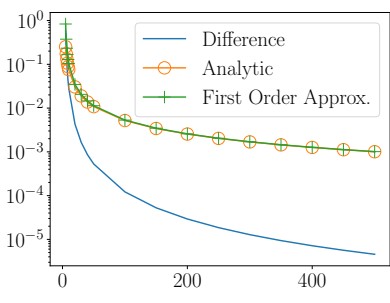

Figure 8: The value of Equation 2, Equation 5 and their difference (absolute difference). The x-axis is $\rho$ and the y-axis is the variance (log scale).

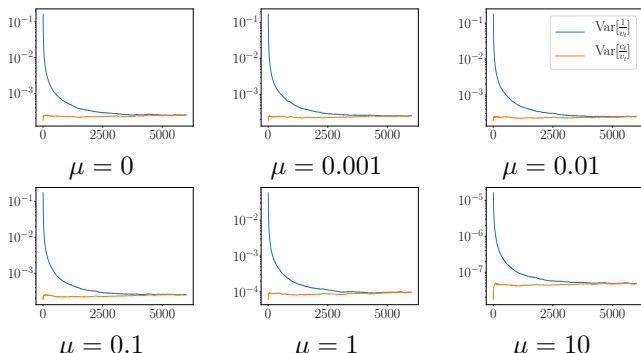

Figure 9: The simulation of $\mathrm{Var}[\frac{1}{v_t}]$ and $\mathrm{Var}[\frac{c_t}{v_t}]$. The x-axis is iteration # (from 5), the y-axis is the variance (log scale).

of Adam, whose adaptive learning rate is rectified so as to have a consistent variance. Empirical results demonstrate the effectiveness of our proposed method. In future work, we plan to replace the rectification strategy by sharing the second moment estimation across similar parameters.

## ACKNOWLEDGE

We thank Aeyuan Allen-Zhu for valuable discussions and comments, Microsoft Research Technology Engineering team for setting up GPU machines. Research was sponsored in part by DARPA No. W911NF-17-C-0099 and FA8750-19-2-1004, National Science Foundation IIS 16-18481, IIS 17-04532, and IIS-17-41317, and DTRA HDTRA11810026.

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

## A  PROOF OF THEOREM 1

For ease of notation, we refer $\psi^2(.)$ as $x$ and $\frac{1}{\sigma^2}$ as $\tau^2$. Thus, $x \sim$ Scale-inv-$\mathcal{X}^2(\rho, \tau^2)$ and:

$$p(x) = \frac{(\tau^2 \rho/2)^{\rho/2}}{\Gamma(\rho/2)} \frac{\exp[\frac{-\rho\tau^2}{2x}]}{x^{1+\rho/2}} \quad \text{and} \quad \mathbb{E}[x] = \frac{\rho}{(\rho - 2)\sigma^2} \ (\forall \, \rho > 2) \tag{6}$$

where $\Gamma(.)$ is the gamma function. Therefore, we have:

$$\mathbb{E}[\sqrt{x}] = \int_0^\infty \sqrt{x}\, p(x)\, dx = \frac{\tau\sqrt{\rho}\,\Gamma((\rho - 1)/2)}{\sqrt{2}\,\Gamma(\rho/2)} \ (\forall \, \rho > 4). \tag{7}$$

Based on Equation 6 and 7, for $\forall \, \rho > 4$, we have:

$$\mathrm{Var}[\psi(.)] = \mathrm{Var}[\sqrt{x}] = \mathbb{E}[x] - \mathbb{E}[\sqrt{x}]^2 = \tau^2\Big(\frac{\rho}{\rho - 2} - \frac{\rho\, 2^{2\rho-5}}{\pi}\mathcal{B}\Big(\frac{\rho - 1}{2}, \frac{\rho - 1}{2}\Big)^2\Big), \tag{8}$$

where $\mathcal{B}(.)$ is the beta function. To prove the monotonic property of $\mathrm{Var}[\psi(.)]$, we need to show:

**Lemma 1.** for $t \geq 4$, $\frac{\partial}{\partial t}\Big(\frac{t}{t-2} - \frac{t\, 2^{2t-5}}{\pi}\mathcal{B}\Big(\frac{t-1}{2}, \frac{t-1}{2}\Big)^2\Big) < 0$

*Proof.* The target inequality can be re-wrote as

$$\frac{\partial}{\partial t}\Big(\frac{t}{t-2} - \frac{t\, 2^{2t-5}}{\pi}\mathcal{B}\Big(\frac{t-1}{2}, \frac{t-1}{2}\Big)^2\Big)$$

$$= \frac{-2}{(t-2)^2} - \frac{2^{2t-5}}{\pi}\mathcal{B}\Big(\frac{t-1}{2}, \frac{t-1}{2}\Big)^2 - \frac{t\, 2^{2t-5}\ln 4}{\pi}\mathcal{B}\Big(\frac{t-1}{2}, \frac{t-1}{2}\Big)^2$$

$$- \frac{2t\, 2^{2t-5}}{\pi}\mathcal{B}\Big(\frac{t-1}{2}, \frac{t-1}{2}\Big)^2\Big(\Psi\Big(\frac{t-1}{2}\Big) - \Psi(t-1)\Big), \quad \Big(\Psi(x) = \frac{\Gamma'(x)}{\Gamma(x)}\Big)$$

$$< 0$$

This inequality is equivalent to:

$$\frac{64\pi}{(t-2)^2 4^t \mathcal{B}(\frac{t-1}{2}, \frac{t-1}{2})^2} + 1 + t\ln 4 + 2t\Psi\Big(\frac{t-1}{2}\Big)$$

$$> 2t\Psi(t-1) \overset{(i)}{=} t\Big[\Psi\Big(\frac{t-1}{2}\Big) + \Psi\Big(\frac{t}{2}\Big) + \ln 4\Big],$$

where $(i)$ is derived from Legendre duplication formula. Simplify the above inequality, we get:

$$\frac{64\pi}{(t-2)^2 4^t \mathcal{B}(\frac{t-1}{2}, \frac{t-1}{2})^2} + 1 + t\Psi\Big(\frac{t-1}{2}\Big) - t\Psi\Big(\frac{t}{2}\Big) > 0,$$

We only need to show

$$\frac{64\pi}{(t-2)^2 4^t \mathcal{B}(\frac{t-1}{2}, \frac{t-1}{2})^2} + 1 + t\Psi\Big(\frac{t-1}{2}\Big) - t\Psi\Big(\frac{t}{2}\Big)$$

$$\geq \frac{64\pi}{(t-2)^2 4^t \mathcal{B}(\frac{t-1}{2}, \frac{t-1}{2})^2} + 2 + t(\ln(t/2) - 1/(t/2 - 0.5)) - t\ln(t/2)$$

$$= \frac{64\pi}{(t-2)^2 4^t \mathcal{B}(\frac{t-1}{2}, \frac{t-1}{2})^2} - \frac{2}{t-1}$$

$$> \frac{64\pi}{(t-2)^2 4^t \mathcal{B}(\frac{t-1}{2}, \frac{t-1}{2})^2} - \frac{2}{t-2} \geq 0,$$

where the first inequality is from $\ln(x) - 1/(2x) > \Psi(x) > \ln(x + 0.5) - 1/x$.

Therefore, we only need to show

$$32\pi \geq (t-2)4^t \mathcal{B}\Big(\frac{t-1}{2}, \frac{t-1}{2}\Big)^2,$$

which is equivalent to

$$(t-2)4^t \mathcal{B}\Big(\frac{t-1}{2}, \frac{t-1}{2}\Big)^2 = (t-2)4^t \frac{\Gamma(\frac{t-1}{2})^4}{\Gamma(t-1)^2}$$

$$\overset{(i)}{=} (t-2)4^t \frac{\Gamma(\frac{t-1}{2})^2}{\Gamma(t/2)^2} 4^{2-t}\pi = 16\pi(t-2)\frac{\Gamma(\frac{t-1}{2})^2}{\Gamma(t/2)^2} \leq 32\pi,$$

where $(i)$ is from Legendre duplication formula.

So we only need to show

$$(t-2)\frac{\Gamma(\frac{t-1}{2})^2}{\Gamma(t/2)^2} \leq 2 \tag{9}$$

Using Gautschi's inequality ($\frac{\Gamma(x+1)}{\Gamma(x+s)} < (x+1)^{1-s}$), we have

$$(t-2)\frac{\Gamma(\frac{t-1}{2})^2}{\Gamma(t/2)^2} \leq (t-2)(\frac{t-1}{2})^{-1} = \frac{2(t-2)}{t-1} < 2 \tag{10}$$

$\square$

## B  Implementation Details

### B.1  Language Modeling

Our implementation is based on the previous work (Liu et al., 2018). Specifically, we use two-layer LSTMs with 2048 hidden states with adaptive softmax to conduct experiments on the one billion words dataset. Word embedding (random initialized) of 300 dimensions is used as the input and the adaptive softmax is incorporated with a default setting (cut-offs are set to $[4000, 40000, 200000]$). Additionally, as pre-processing, we replace all tokens occurring equal or less than 3 times with as UNK, which shrinks the dictionary from 7.9M to 6.4M. Dropout is applied to each layer with a ratio of 0.1, gradients are clipped at 5.0. We use the default hyper-parameters to update moving averages, $i.e.\beta_1 = 0.9$ and $\beta_2 = 0.999$. The learning rate is set to start from 0.001, and decayed at the start of 10th epochs. LSTMs are unrolled for 20 steps without resetting the LSTM states and the batch size is set to 128. All models are trained on one NVIDIA Tesla V100 GPU.

### B.2  Imageine Classification

We use the default ResNet architectures (He et al., 2016) in a public pytorch re-implementation[4]. Specifically, we use 20-layer ResNet (9 Basic Blocks) for CIFAR-10 and 18-layer ResNet (8 Basic Blocks) for ImageNet. Batch size is 128 for CIFAR-10 and 256 for ImageNet. The model is trained for 186 epoches and the learning rate decays at the 81-th and the 122-th epoches by 0.1 on CIFAR-10, while the model is trained for 90 epoches and the learning rate decays at the 31-th and the 61-th epoch by 0.1 on ImageNet. For Adam and RAdam, we set $\beta_1 = 0.9, \beta_2 = 0.999$. For SGD, we set the momentum factor as 0.9. The weight decay rate is $10^{-4}$. Random cropping and random horizontal flipping are applied to training data.

### B.3  Neural Machine Translation

Our experiments are based on the default Transformers (Vaswani et al., 2017) implementation from the fairseq package (Ott et al., 2019). Specifically, we use word embedding with 512 dimensions and 6-layer encoder / decoder with 4 head and 1024 hidden dimensions on the IWSLT14' dataset; use word embedding with 512 dimension and 6-layer encoder / decoder with 8 heads and 2048 hidden dimensions. Label smoothed cross entropy is used as the objective function with an uncertainty $= 0.1$ (Szegedy et al., 2016). We use linear learning rate decay starting from $3e^{-4}$, and the checkpoints of the last 20 epoches are averaged before evaluation. As to the wamrup strategy, we use a linear warmup for Adam in the first $4000$ updates, and set $\beta_2$ to satisfy $\nu = 4000$ ($\beta_2 = 0.9995$). In the IWSLT'14 dataset, we conduct training on one NVIDIA Tesla V100 GPU, set maximum batch size as 4000, apply dropout with a ratio 0.3, using weight decay of 0.0001 and clip the gradient norm at 25. In the WMT'16 dataset, we conduct training on four NVIDIA Quadro R8000 GPUs and set maximum batch size as 8196.

## C  Downgrading to SGDM

As a byproduct determined by math derivations, we degenerated RAdam to SGD with momentum in the first several updates. Although this stage only contains several gradient updates, these up-

---

[4]https://github.com/bearpaw/pytorch-classification

dates could be quite damaging (e.g., in our Figure 2, the gradient distribution is distorted within 10 gradient updates). Intuitively, updates with divergent adaptive learning rate variance could be more damaging than the ones with converged variance, as divergent variance implies more instability. As a case study, we performed experiments on the CIFAR10 dataset. Five-run average results are summarized in Table 3. The optimizer fails to get an equally reliably model when changing the first 4 updates to Adam, yet the influence of switching is less deleterious when we change 5-8 updates instead. This result verifies our intuition and is in agreement with our theory  the first few updates could be more damaging than later updates. By saying that, we still want to emphasize that this part (downgrading to SGDM) is only a minor part of our algorithm design whereas our main focus is on the mechanism of warmup and the derivation of the rectification term.

Table 3: Performance on CIFAR10 (lr = 0.1).

| 1-4 steps | 5-8 steps | 8+ steps | test acc | train loss | train error |
|---|---|---|---|---|---|
| RAdam | RAdam | RAdam | 91.08 | 0.021 | 0.74 |
| Adam (w. divergent var.) | RAdam | RAdam | 89.98 | 0.060 | 2.12 |
| SGD | Adam (w. convergent var.) | RAdam | 90.29 | 0.038 | 1.23 |

