# OpenReview forum: "On the Variance of the Adaptive Learning Rate and Beyond"
_ICLR.cc/2020/Conference — Accept (Poster)_

### Official Review · AnonReviewer3 · 2019-10-23
**Official Blind Review #3**

**Rating:** 6

**Review:**

Authors propose a way to rectify the variance of the adaptive learning rate (RAdam) and apply the optimizer to applications in image classification, language modeling and neural machine translation. The experiments demonstrate not only a strong results over baseline Adam with warmup learning rate but the robustness of the optimizer. The authors additionally demonstrate the theoretical justification behind their optimizer, however I am not very qualified to make the judgement on the theory. Overall judging from the authors description of approach and experimental results, I recommend acceptance.


**Experience Assessment:**

I have published in this field for several years.

**Review Assessment: Checking Correctness Of Derivations And Theory:**

I did not assess the derivations or theory.

**Review Assessment: Checking Correctness Of Experiments:**

I assessed the sensibility of the experiments.

**Review Assessment: Thoroughness In Paper Reading:**

I read the paper at least twice and used my best judgement in assessing the paper.

---

> ### Author Response · Authors · 2019-11-10
> **Re: Official Blind Review #3**
>
> Thank you for your review and feedback.
>
> Although the warmup technology has been demonstrated to be useful in several applications and domains, it is not regarded as a common practice, partially because it is unclear why we need such technologies. In this study, we aim to uncover its underpinnings and identify an important yet long-overlooked issue: the adaptive learning rate has a problematically large variance in the early stage of training, due to the lack of enough samples. Based on our analysis, we further propose a rectification term to address this issue. In our experiments, we show that it works well on various tasks/domains.

---

### Official Review · AnonReviewer1 · 2019-10-24
**Official Blind Review #1**

**Rating:** 6

**Review:**

I haven't worked in this area before, and my knowledge to the topic of designing optimizers for improving stochastic gradient descent is limited to the level of advanced ML courses at graduate school. Nevertheless, below I try my best to evaluate the technicality of this paper

====================
In this work the authors studied the variance issue of the adaptive learning rate and aim to justify the warm-start heuristic. They also demonstrate that  the convergence issue of many of the stochastic gradient descent algorithms is due to large variance induced by the adaptive learning rate in the early stage of training. To tackle this issue, they proposed a variant of ADAM, which is known as rectified ADAM, whose learning rate not only takes the momentum into the account, but it also adapts to the variance of the previous gradient updates.

In the first part of the paper, the authors analyzed the variance issue exists in the existing ADAM algorithm, such that with limited samples in the early stage of training, the variance of the adaptive learning rate becomes rather large and it induces high variance to the gradient update to ADAM. In general I found this theoretical justification on the observation of variance issue in ADAM sound, and quite intuitive. In the second part, they proposed the algorithm, namely rectified ADAM, where the difference here to take the second moment  of the gradient into account when updating the adaptive learning rate. They showed that the variance of the adaptive learning rate with such rectification is more numerically stable (especially when variance is intractable in vanilla ADAM), and under some regularity assumption it decreases in the order or O(1/\rho_t).

In extensive numerical studies of supervised learning, the authors showed that RADAM achieves a better accuracy than ADAM (although in Table 1, I am a bit puzzled why the best accuracy is indeed from SGD, if so, what's the point of all adaptive learning rates, is that because SGD requires extensive lr tuning?) Because the superiority in accuracy they also showed that RADAM manages to have more stable training and achieves lower training loss, which is quite interesting.


**Experience Assessment:**

I do not know much about this area.

**Review Assessment: Checking Correctness Of Derivations And Theory:**

I assessed the sensibility of the derivations and theory.

**Review Assessment: Checking Correctness Of Experiments:**

I assessed the sensibility of the experiments.

**Review Assessment: Thoroughness In Paper Reading:**

I read the paper at least twice and used my best judgement in assessing the paper.

---

> ### Author Response · Authors · 2019-11-10
> **Re: Official Blind Review #1**
>
> Thank you for your review and feedback. We understand your concern and hope that our response will alleviate it. More specifically,
>
> > On the comparison between SGDM and adaptive optimization algorithms
>
> It is true that for datasets like ImageNet, the state of the art resnet performance is usually achieved by SGD with momentum (SGDM). In our case, we observe a similar phenomenon, and we suggest it because the hyper-parameters are tuned for the SGDM and may be sub-optimal for other algorithms.
>
> Comparing to SGDM, the adaptive optimization algorithms usually converge faster and are more robust to the choice of hyper-parameters, thus have been viewed as the default choice in many applications [1,2]. Based on our experience, it requires non-trivial efforts to make SGDM achieve similar performance for cases like training Transformers.
>
> 1. Reimers, Nils, and Iryna Gurevych. "Optimal hyperparameters for deep lstm-networks for sequence labeling tasks." arXiv preprint arXiv:1707.06799 (2017).
> 2. Popel, Martin, and Ondřej Bojar. "Training tips for the transformer model." The Prague Bulletin of Mathematical Linguistics 110.1 (2018): 43-70.

---

### Official Review · AnonReviewer2 · 2019-10-26
**Official Blind Review #2**

**Rating:** 6

**Review:**

In this work, authors show that the bad performance of Adam is from the large variance of adaptive learning rate at the beginning of the training.
Pros:
1.	Authors demonstrate that the variance of the first few stages is large, which may interpret the degradation in the performance of Adam.
2.	The empirical study supports the claim about the large variance.

Cons:
1.	Theoretically, authors didn’t illustrate why the large variance can result in the bad performance in terms of, e.g., convergence rate, generalization error, etc.
2.	The performance of the proposed algorithm is still worse than SGD and it makes the analysis less attractive.


**Experience Assessment:**

I have read many papers in this area.

**Review Assessment: Checking Correctness Of Derivations And Theory:**

I assessed the sensibility of the derivations and theory.

**Review Assessment: Checking Correctness Of Experiments:**

I assessed the sensibility of the experiments.

**Review Assessment: Thoroughness In Paper Reading:**

I read the paper at least twice and used my best judgement in assessing the paper.

---

> ### Author Response · Authors · 2019-11-10
> **Re: Official Blind Review #2**
>
> Thank you for your review and feedback. We understand your concern and hope that our response will alleviate them. More specifically,
>
> > On theoretic analysis of the impact of the large variance
>
> In our study, we focus on exploring the underlying mechanism of the warmup technology, and find that it is non-trivial to theoretically identify the existence of the variance issue. Due to the complicated nature of neural networks, we believe that it would be even more challenging to establish a general and direct connection between large variance and the neural network behavior — in fact, a theoretical analysis of the neural network behavior by itself is a big challenge. Therefore, we believe that these questions deserve a more in-depth analysis, and we would like to leave it to future work.
> Here, we want to borrow some insights from recently proposed theories to intuitively illustrate why large variance of the adaptive learning rate is harmful in a simplified case. It is worth mentioning that these results are based on simple model structures (e.g. two-layer CNN/ResNet) and strong data distribution assumptions [1, 2, 3].
>
> It has been shown that there are bad local optima when optimizing neural networks [2, 3]; and it requires the learning rate of the gradient descent algorithm (not SGD) to be controlled within a range, in order to avoid being trapped in the regions of bad local optima and converging to the global optimum [2, 3]. In other words, the learning rate cannot be too large and has to be set within a range. Such condition might be compromised by the large variance of the adaptive learning rate — as variance is a measure of variability (i.e., lack of consistency or fixed pattern), the adaptive learning rate with a larger variance is less likely to be held within the desired range.
>
> 1. Ge, Rong, et al. "Learning two-layer neural networks with symmetric inputs." International Conference on Learning Representations (ICLR), 2019.
> 2. Du, Simon S., et al. "Gradient descent learns one-hidden-layer cnn: Don't be afraid of spurious local minima." International Conference on Machine Learning (ICML) 2019.
> 3. Liu, Tianyi, et al. "Towards Understanding the Importance of Shortcut Connections in Residual Networks."Annual Conference on Neural Information Processing Systems (NeurIPS), 2019.
>
> > On the accuracy of SGDM and adaptive optimization algorithms
>
> It is true that for datasets like ImageNet, the state of the art resnet performance is usually achieved by SGD with momentum (SGDM). In our case, we observe a similar phenomenon, and we suggest it because the hyper-parameters are tuned for SGDM and may be sub-optimal for other algorithms. At the same time, our proposed RAdam algorithm shows more robustness towards learning rate changes. It is also worth mentioning that, although RAdam fails to outperform SGD in terms of test accuracy, it results in faster convergence, lower training loss and better training performance (e.g., the training accuracy of SGD, Adam, and RAdam on ImageNet are 69.57, 69.12 and 70.30 respectively).

---

### Public Comment · ~Boris_Ginsburg1 · 2019-10-06
**Source of instability during initial training phase?**

The main claim of the paper is that "the lack of sufficient data samples in the early stage is the root cause of the convergence issue" (section 3.1). How you would explain the following observations:
1. Adam works well with  small batch size without warmup. Why in this case the small amount of samples doesn't impact the stability?
2. Learning rate warm-up was introduced for large batch training. large batch usually have thousands of samples. Then what is the source  of instability during initial training phase?

Btw, there is a typo in Algorithm 1, line 5: v_t instead of l_t

---

> ### Author Response · Authors · 2019-10-06
> **Although warmup helps both Adam and SGD, they are for different reasons.**
>
> Thanks for asking and pointing out the typo, we will fix it in the next version. : -)
>
> 1. Our study is motivated by the phenomenon that: Adam-without-warmup fails for NMT, even with small batch size. As in Theorem 1, the adaptive learning rate has a larger variance in the early stage of training than the later stage. Also, by providing two thousand additional batches to estimate the learning rate, we can avoid the convergence problem met by the vanilla-Adam.
>
> 2. We agree the warmup is originally designed for large-minibatch-SGD [0], based on the intuition that the network changes rapidly in the early stage. However, we find that this intuition does not explain why Adam requires warmup. Notice that, Adam-2k can also avoid the convergence problem: it uses the same learning rate and initialization but has a better estimation of the adaptive learning rate (more details below). Therefore, we suggest that although warmup helps both Adam and SGD, they are for different reasons.
>
> In the first two thousand batches, Adam-2k will only update the moving average of the second moment, and freeze both the first moment and the parameter value; after these two thousand batches, Adam-2k will start to update parameters in the vanilla-Adam way. That is to say, Adam-2k (avoids convergence problem) and Adam (suffers from convergence problem) has the same initialization, learning rate scheduler and update rule; their only difference is Adam-2k has additional samples to estimate the adaptive learning rate. Accordingly, we think the root cause of the convergence issue is the lack of sufficient data samples in the early stage (to estimate the adaptive learning rate).
>
> The reason why sometimes warmup also helps SGD still lacks theoretical support. We believe this topic deserves more in-depth analysis and is beyond the scope of this study.
>
> [0] Goyal et al, Accurate, Large Minibatch SGD: Training Imagenet in 1 Hour, 2017

---

> > ### Public Comment · ~Boris_Ginsburg1 · 2019-10-06
> > **Why does warmup help?**
> >
> > Question: If you  would combine all samples from first 2K iterations into one large batch, would you still need warmup?
> >
> > "The reason why sometimes warmup also helps SGD still lacks theoretical support. We believe this topic deserves more in-depth analysis and is beyond the scope of this study. "
> > One explanation is that during initial phase, the main issue is not number of observed samples, but the fact that the  update is much larger than weights (LARS, LAMB, ....).

---

> > > ### Author Response · Authors · 2019-10-07
> > > **Re: Why does warmup help**
> > >
> > > Thanks for asking. I did an experiment and it shows: treating the first 2001 batches as one large batch suffers from the same convergence problem with the vanilla-Adam (in our NMT experiment as in Figure 1). It further verifies our hypothesis that the adaptive learning rate has undesirably large variance and blocks the algorithm from getting a reasonable performance.
> > >
> > > Empirically, improper learning rate setting can lead to convergence problems. Still, to explain the underlying mechanism of warmup, it requires a detailed analysis on the difference between the initial phrase and the later phrase (e.g., why it requires a smaller learning rate in the beginning; or with the same update rule and learning rate, why the update is more problematic in the beginning).
> > >
> > > Based on our experiments, after getting enough samples to estimate the adaptive learning rate (instead of the gradient direction), we can avoid the convergence problem as in Figure 1.
> > >
> > > We will explore the problem why warmup helps SGD in the future, and thanks for pointing out one potential explanation.

---

> > > > ### Public Comment · ~Boris_Ginsburg1 · 2019-10-07
> > > > **Adam-2K**
> > > >
> > > > "In the first two thousand batches, Adam-2k will only update the moving average of the second moment, and freeze both the first moment and the parameter value; after these two thousand batches, Adam-2k will start to update parameters in the vanilla-Adam way."
> > > >
> > > > When you say that parameters are frozen for first 2000 iteration, does it mean that first 2000 iterations you compute gradients for the same weights, and that you use these gradients only to update second momentum without updating first moment and weights?

---

> > > > > ### Author Response · Authors · 2019-10-07
> > > > > **Re: Adam-2k**
> > > > >
> > > > > Yes, your understanding is correct.

---

### Author Response · Authors · 2019-10-06
**Corrections of Typos**

We've found several typos (i.e., in Algorithm 1, line 5, Equation 4 and Equation 7). Sorry for the mistake and the corrected version can be found in: https://drive.google.com/open?id=1UQbRm66IMPP8_HycUIP-txV2vNx9SneT

---

### Public Comment · ~Jerry_Ma1 · 2019-11-06
**Public comments on the present manuscript**

Hello,

We would like to provide a couple of comments about this present manuscript proposing the RAdam algorithm. These comments are based on our work in shorturl.at/uyzY8 (suitably edited to preserve anonymity of the manuscript’s authors).

On the "divergent variance" correction:

* RAdam is 4 timesteps of momentum SGD, followed by Adam with a fixed warmup schedule. For all values of $\beta_2$ usable in practice, the condition "if the variance is tractable, i.e., $\rho_t = \rho_{\infty} - 2 t \beta_2^t / (1 - \beta_2^t) > 4$" (Algorithm 2, Line 9) is precisely equivalent to the condition "if $t > 4$". Neglecting to state this lends an air of “smoke and mirrors” by obfuscating the true operation of the algorithm under the imprimatur of mathematical sophistication.

* The manuscript does not demonstrate the necessity of 4 steps of momentum SGD on any task. Indeed, we were unable to find any realistic setting where 4 steps of momentum SGD ended up doing anything remotely discernible. In the absence of any practical justification of an arbitrary 4 steps of momentum, we view the "divergent variance" correction as magic masquerading as math.

Ignoring the 4 steps of momentum SGD, RAdam is precisely Adam with a fixed learning rate warmup schedule. This raises the following points:

* The manuscript attempts to separate RAdam from standard "heuristic" warmup schedules in both motivation and operation. In truth, RAdam without the 4 steps of momentum SGD is a specific functional form for heuristic warmup that depends on $\beta_2$.

* We found that RAdam performs identically to linear warmup over $2 / (1 - \beta_2)$ timesteps, across all domains evaluated in the manuscript.

* The manuscript claims as RAdam’s primary benefit that no manual tuning of a warmup schedule is needed for robust operation. To the extent that this claim is true, it is also true of linear warmup given the above.

In our opinion, a complex method should not be advanced over a simpler method unless the complex method brings something useful to the table. There is no evidence that RAdam brings anything to the table above and beyond the simplest of warmup methods, such as linear warmup over $2 / (1 - \beta_2)$ timesteps.

Finally, we would like to point out the following about the underlying motivation for RAdam and learning rate warmup:

* The manuscript does not show a causal relationship between the variance of the adaptive learning rate and training instability, nor does it show any significant consequence of "divergent variance" versus “convergent variance” (i.e. comparing timestep 4 and timestep 5) – probably because there are none.

* While it may be true that the variance of the adaptive learning rate (i.e. inverse second moments) is /correlated/ with training instability, any causal relationship to training instability is likely with the parameter update magnitudes; understanding the latter requires jointly analyzing the first moments and second moments, as they are extremely dependent on each other.

In summary, we believe that endorsing this manuscript in present form with inclusion at ICLR will encourage incremental and poorly-justified future work in new Adam variants and learning rate warmup.

- Jerry Ma and Denis Yarats

---

> ### Author Response · Authors · 2019-11-10
> **Official Response (2/2)**
>
> > **Math Derivations**
>
> One purpose of our study is to rigorously explore the underpinning of warmup — we believe that only functioning as the outcome of math derivations, our method can explicitly handle the variance issue. However, the comment criticizes math derivations as insufficient to justify our algorithm design, referring to them as “magic masquerading” and “smoke and mirrors”.
>
> We respectfully disagree with this statement, as math is an important and powerful tool to formulate and verify our understanding. We believe in order to perform a rigorous analysis, math derivations are necessary and helpful.
>
> In our study, we conduct theoretical analysis besides controlled experiments and find that math agrees with our hypothesis (it is the large variance caused the training instability). Specifically, our theory shows that, at the beginning of training, the variance of the adaptive learning rate can be undesirably large, verifying the existence of the variance issue. Additionally, this inspires us to introduce a mathematically sound rectification term to handle the variance.
>
> > **Downgrading to SGDM**
>
> A large portion of the comment is about the issue of the RAdam downgrading (i.e.,"4 timesteps of momentum SGD"), a byproduct determined by math derivations. Although this is not the main result of our paper, we'd like to clarify why it is designed this way below.
>
> Here, we find the troublesome large variance of the adaptive learning rate can cause training instability (thus we focus on the magnitude instead of the divergence). From our theoretical analysis of the adaptive learning rate variance, we derive the rectification term to handle this issue. However, we are constrained from applying such a rectification term at the very beginning of training, since the variance of the estimated adaptive learning rate is not well-defined (in other words, divergent). Therefore, we downgrade the algorithm to SGDM in this stage.
>
> Although this stage only contains several gradient updates, these updates could be quite damaging (e.g., in our Figure 2, the gradient distribution is distorted within 10 gradient updates). Intuitively, updates with divergent adaptive learning rate variance could be more damaging than the ones with converged variance, as divergent variance implies more instability. As a case study, we performed experiments on the CIFAR10 dataset. Five-run average results are summarized below. The optimizer fails to get an equally reliably model when changing the first 4 updates to Adam, yet the influence of switching is less deleterious when we change 5-8 updates instead. This result verifies our intuition and is in agreement with our theory — the first few updates could be more damaging than later updates. By saying that, we still want to emphasize that this part (downgrading to SGDM) is only a minor part of our algorithm design whereas our main focus is on the mechanism of warmup and the derivation of the rectification term.
>
> +--------------------------------------------------------------------------------------------------------------------------------------------------------+
> |                                                                            Performance on CIFAR10 (lr = 0.1)                                                              |
> +-----------------------------------------------------------------------------------------------------+-------------+---------------+------------------+
> |                                                                                                                                    |   test acc |  train loss |   train error |
> +-----------------------------------------------------------------------------------------------------+-------------+---------------+------------------+
> | All steps: RAdam                                                                                                     |       91.08 |         0.021 |             0.74  |
> +-----------------------------------------------------------------------------------------------------+-------------+---------------+------------------+
> | 1-4 steps (divergent variance): Adam; 5+ steps: RAdam                                 |      89.98 |          0.060 |             2.12  |
> +-----------------------------------------------------------------------------------------------------+-------------+---------------+------------------+
> | 1-4 steps: SGD; 5-8 steps (convergent variance): Adam; 8+ steps: Radam  |      90.29 |         0.038 |              1.34  |
> +-----------------------------------------------------------------------------------------------------+-------------+---------------+------------------+

---

> ### Author Response · Authors · 2019-11-10
> **Official Response (1/2)**
>
> The comment summarizes our work as “incremental and poorly-justified”, mainly based on the similarity between RAdam and two warmup schedulers; however, we find that such similarity ends up supporting our intuition — the needs of warmup comes from the problematic adaptive learning rate, since these two schedulers are entirely controlled by $\beta_2$ (the only hyper-parameter for calculating the adaptive learning rate).
>
> Before we address the comment in detail, we want to point out that the main focus of our study is on exploring the underlying mechanism of warmup. Our analysis reveals that the training instability (at least in our NMT case) is mainly caused by adaptive learning rate, and warmup is needed to handle the large variance of the adaptive learning rate at the early stage of training. Inspired by the analysis, we propose a new variant of Adam by introducing a rectification term to explicitly control the variance of the adaptive learning rate.
>
> > **Comparing with Heuristic LR Scheduler**
>
> In their manuscript, it says “RAdam and the untuned rule-of-thumb warmup schedules are more or less interchangeable”, thus “RAdam: perform 4 iterations of momentum SGD, then use Adam with fixed warmup”.
>
> However, since the fixed warmup schedulers (in the above argument) are controlled solely by the choice of $\beta_2$, only the adaptive learning rate is taken into consideration. Therefore, their designs are based on an intuition similar to ours — in the first stage of training, due to the use of insufficient samples for estimating the adaptive learning rate, a warmup stage is needed (and should be customized via the choice of \beta_2). Also, as shown in Figure 3 of the shared manuscript, the compared lr scheduler is very similar to our current design; it is not a surprise to find that their performances are also similar.
>
> To put it differently, we think that this phenomenon supports our intuition on the relationship between training instability and the lack of enough samples for adaptive learning rate estimation.
>
> > **Adaptive Learning Rate Variance**
>
> In the comment, it says “The manuscript does not show a causal relationship between the variance of the adaptive learning rate and training instability.”
>
> Variance is a measure of variability, i.e., lack of consistency or fixed pattern. Therefore, we pick variance as the measure to study stability [1]. Intuitively, if the adaptive learning rate is not stable, it will have a large variance.
>
> To seek the origin of training instability, we designed two controlled experiments (Adam-2k and Adam-eps; see Sec 3.1). Specifically, Adam-2k narrows down the problem to the adaptive learning rate, and Adam-eps shows the convergent problem can be avoided by stabilizing the adaptive learning rate (large eps can be viewed as a small-gradient filter). Accordingly, these experimental results suggest that the adaptive learning rate’s instability (large variance) is a major, if not the only, contributor to training instability.
>
> [1] Beard R.E., Pentikäinen T., Pesonen E. (1984) Variance as a measure of stability. In: Risk Theory. Monographs on Statistics and Applied Probability, vol 20. Springer, Dordrecht

---

### Decision · Program_Chairs · 2019-12-19

**Decision:**

Accept (Poster)

**Comment:**

The paper considers an important topic of the warmup in deep learning, and investigates the problem of the adaptive learning rate. While the paper is somewhat borderline, the reviewers agree that it might be useful to present it to the  ICLR community.